# SymTFT construction of
# gapless exotic-foliated dual models

Fabio Apruzzi[1], Francesco Bedogna[1], Salvo Mancani[1]

[1] *Dipartimento di Fisica e Astronomia "Galileo Galilei", Università di Padova,*
*Via Marzolo 8, 35131 Padova, Italy*

[1] *INFN, Sezione di Padova Via Marzolo 8, 35131 Padova, Italy*

We construct Symmetry Topological Field Theories (SymTFTs) for continuous subsystem symmetries, which are inherently non-Lorentz-invariant. Our framework produces dual bulk descriptions—gapped foliated and exotic SymTFTs—that generates gapless boundary theories with spontaneous subsystem symmetry breaking via interval compactification. In analogy with the sandwich construction of SymTFT, we call this *Mille-feuille*. This is done by specifying gapped and symmetry-breaking boundary conditions. In this way we obtain the foliated dual realizations of various models, including the XY plaquette, XYZ cube, and $\phi$, $\hat{\phi}$ theories. This also captures self-duality symmetries as condensation defects and provides a systematic method for generating free theories that non-linearly realize subsystem symmetries.

# 1  Introduction

In recent years, the notion of symmetry in quantum field theory has undergone a profound generalization, extending far beyond the traditional framework of global, continuous, and invertible symmetries [1]. This broader perspective encompasses generalized symmetries (see [2–8] for a review), which include higher-form symmetries associated with extended operators, noninvertible symmetries, and subsystem symmetries that act along lower-dimensional manifolds such as lines or planes within the system. These diverse symmetry structures have proven indispensable in characterizing novel quantum phases of matter, particularly in strongly correlated systems and in the presence of topological order. The interplay between such symmetries not only enriches the classification of field theories but also guides the construction of effective models and dualities, especially in settings where conventional symmetry descriptions fail.

A powerful framework for encoding generalized symmetries is provided by the Symmetry Topological Field Theory (SymTFT) construction. SymTFTs are topological field theories that encapsulate the universal symmetry properties of a given quantum field theory (QFT). This framework has been extensively developed for finite symmetries [1, 3, 9–20], and more recently extended to continuous symmetries [21–24] as well as finite subsystem and modulated symmetries [25, 26].

Given a $d$-dimensional QFT $\mathcal{T}$ with symmetry $G$, one can construct a $(d+1)$-dimensional TQFT with gauge group $G$, whose gauge fields couple to $\mathcal{T}$, which resides on a $d$-dimensional boundary. We refer to this TQFT as the SymTFT, and to the boundary supporting the QFT as the physical boundary, $\mathcal{B}^{\text{phys}}$. The topological operators of the SymTFT give rise to symmetry operators of the boundary QFT and/or source charged operators when they are projected onto or terminate at the boundary. To fully recover the QFT $\mathcal{T}$, one must also specify a topological boundary condition on a second, distinguished boundary. This places the SymTFT on a spacetime of the form $M_{d+1} = M_d \times I$, where $I$ is a finite interval. Upon compactifying the interval direction, one recovers the QFT $\mathcal{T}$ with symmetry $G$. Varying the topological boundary condition alters both the resulting QFT and its associated symmetry $G$—a procedure known as the *sandwich* construction. For instance, in pure gauge theories with 1-form symmetries and no matter content, one can uniformly describe all global variants within a single SymTFT by varying the topological boundary condition. This framework applies to both finite and continuous symmetry groups.

A natural extension of this framework involves *subsystem symmetries*, which act not globally but along lower-dimensional submanifolds—such as lines or planes—in space. These symmetries arise prominently in fracton phases and other models with constrained mobility

on the lattice and in the continuum, and they fall outside the traditional classification of global or higher-form symmetries [27–45, 45–63]. Unlike conventional symmetries, subsystem symmetries are intrinsically tied to the geometry of the system, often leading to exotic behavior such as ground state degeneracy and restricted operator dynamics. Recent work [25, 26] has shown that SymTFT constructions can be adapted to incorporate such symmetries by replacing the standard bulk gauge theory with more intricate foliated or modulated structures that encode the direction-dependent nature of subsystem transformations [42, 43]. This opens the door to a unified treatment of foliated field theories, fracton dualities, and gapless phases enriched by subsystem symmetry, within a generalized topological framework.

In this work, we construct the SymTFT for continuous subsystem symmetries, which are inherently non-Lorentz-invariant. We propose two dual bulk SymTFTs that capture the same symmetry structure: a gapped foliated theory and an exotic gapped theory, each serving as a distinct bulk SymTFT for the same boundary physics. Our central goal is to construct gapless foliated models that spontaneously break the subsystem symmetry under study in the continuum. This is achieved by specifying two distinct boundary conditions: a gapped boundary condition and a spontaneous symmetry-breaking (ssb) boundary condition, called also singleton [64] or scale/conformal invariant boundary condition. Upon compactifying the interval direction, this setup yields dual descriptions of gapless phases with spontaneously broken subsystem symmetry [65, 66]. This construction is analogous to the sandwich one, but due to the foliated nature of the bulk, and following the naming tradition of SymTFT sandwich constructions, we dub this setup as *Mille-feuille*.

Notably, using the exotic SymTFT, the mille-feuille recovers the Lagrangians of models previously studied in [59]. Crucially the foliated SymTFT gives rise to dual gapless foliated models. This is in constrast with the foliated models related to the exotic by RG-flows [55]. We demonstrate the power of this construction through explicit examples, including the XY-plaquette and and XYZ-cube with continuous subsystem symmetry, as well as the $\phi$ and $\hat{\phi}$ models [42, 43, 45]. Furthermore, we provide an example for a self-duality defect as condensation defects in the SymTFTs.

Finally, this work provides a systematic way to construct free gapless theories that non-linearly realize continuous subsystem symmetries. By using the SymTFT framework with appropriate bulk and boundary data, we can generate a wide class of models that exhibit spontaneous subsystem symmetry breaking, along with their dual descriptions.

The paper is organized as follows. In section 2 we review the SymTFT for lorentz invariant continuous symmetries and how to construct gapless models that non non-linearly realize a $U(1)$ symmetry focusing on the 2-dimensional case [65]. In section 3 we introduce subsystem

symmetries and continuous largangian models that have them. In section 4 we provide the non-Lorentz invariant SymTFT for subsystem symmetries both in its exotic and foliated versionBy using an analogous strategy as the one highlighted in 2, we construct the foliated gapless models dual to the one of section 3.

**Note added:** The authors thank K. Ohmori for sharing the information that the work [67], which has some overlap with the results of this paper, is in preparation.

## 2   SymTFT for continuous abelian symmetries and SSB

SymTFTs for continuous abelian symmetries have been introduced in [21,22]. Symmetry properties of $U(1)$ p-form symmetries in $d$-dimension can be captured by the following SymTFT, [1]

$$S_{d+1} = \frac{1}{2\pi} \int_{M_{d+1}} b_{d-p-1} \wedge dc_{p+1} \tag{2.1}$$

where $b_{d-p-1}$ and $c_{p+1}$ are $\mathbb{R}$-valued gauge field, i.e. their gauge transformation read,

$$c_{p+1} \to c_{p+1} + d\lambda^{(c)}_{p+1}, \quad b_{d-p-1} \to b_{d-p-1} + d\lambda^{(b)}_{d-p-1}. \tag{2.2}$$

and there are no large gauge transformation. The equation of motion imply that $dc_{p+1} = db_{d-p-1} = 0$. The topological operators of this theory are given by

$$V_x(\Sigma_{p+1}) = e^{ix \int_{\Sigma_{p+1}} c_{p+1}}, \quad U_y(\Sigma_{d-p-1}) = e^{iy \int_{\Sigma_{d-p-1}} b_{d-p-1}}. \tag{2.3}$$

where $x, y \in .\mathbb{R}$. We can now compute how $U_y, V_x$ act on each other by evaluating the following correlator,

$$\langle U_y(\Sigma_{d-p-1}), V_x(\Sigma_{p+1}) \rangle = \exp(i\text{Link}(\Sigma_{d-p-1}, \Sigma_{p+1})) \tag{2.4}$$

which can be seen by inserting the topological defects in the action as sources, by solving the new equation of motions and integrating out the gauge field $c_{p+1}, b_{d-p-1}$.

We now describe the SymTFT sandwich construction and we specialize to $d = 2$ and $p = 1$, so that $b_1 = b$ and $c_1 = c$ are two $\mathbb{R}$-valued 1-form gauge fields. In particular we review the strategy highlighted in [65], implemented to derive topological manipulations of the 2d compact boson theory, which non-linearly realizes the $U(1)$ 0-form symmetry described by the SymTFT in the bulk. This is done by changing the topological (gapped) boundary condition. The SymTFT action reads

$$S = \frac{1}{2\pi} \int_{M_3} b \wedge dc . \tag{2.5}$$

---

[1] Any coefficient different from 1 can be reabsorbed in a redefinition of gauge field.

In order to realize the sandwich picture of SymTFT we have to place the theory in a space $M_3 = I_{[0,L]} \times M_2$ with two boundaries $M_2^{(0)}$ and $M_2^{(L)}$. At $M_2^{(0)}$ we will impose topological boundary conditions, given by adding the following boundary action

$$S_2^{(0)} = \frac{iR}{2\pi} \int_{M_2^{(0)}} \phi dc \tag{2.6}$$

where $\phi$ lives only in $M_2^{(0)}$ and it is a periodic $U(1)$ scalar, that transforms also as follows,

$$\phi \to \phi - R^{-1}\lambda^{(b)} \tag{2.7}$$

such that the bulk gauge variation

$$\delta_{\lambda^{(b)}} S = \int_{M_2^{(0)}} \lambda^{(b)} dc \tag{2.8}$$

is cancelled by (2.7).

This boundary action will set which defects end on the boundary and which one get parallel projected. For instance the boundary variational problem sets,

$$b\big|_{M_2^{(0)}} = -Rd\phi . \tag{2.9}$$

In addition since $\phi$ is periodic and $\oint d\phi = 2\pi\mathbb{Z}$, the integration over these configurations sets

$$\int_{\gamma_1 \subset M_2^{(0)}} c \in 2\pi R^{-1}\mathbb{Z}. \tag{2.10}$$

This topological boundary condition allows the the topological operators $V_{Rn}, U_{\frac{n}{R}}$ to end on the boundary, which implies that the following identification for the operators (2.4) when projected on $M_2^{(0)}$,

$$x \sim x + \frac{1}{R}, \quad y \sim y + R \tag{2.11}$$

hence defining the $U(1) \times U(1)$ symmetry.

## 2.1 Sandwich construction with conformal boundary condition

We are interested in imposing a physical boundary conditions on the other boundary $M_2^{(L)}$, that realizes the bulk symmetry non-linearly via edge modes, without adding any additional degrees of freedom at the boundary. This boundary condition entails adding the following singleton term [22, 64, 65, 68–72]

$$S_2^{(L)} = \frac{1}{4\pi} \int_{M_2^{(L)}} c \wedge \star_2 c, \tag{2.12}$$

The variational problem sets

$$b\big|_{M_2^{(L)}} = i \star_2 c\big|_{M_2^{(L)}} . \tag{2.13}$$

We can now compactify the interval direction $L \to 0$. This is done by integrating out the bulk and the boundary action which has contributions from (2.12) and (2.6). Imposing the boundary conditions (2.9) and (2.13) we get the following theory

$$S_2 = \frac{iR}{4\pi} \int_{M_2} d\phi \wedge \star_2 d\phi \tag{2.14}$$

where $M_2 = M_2^{(0)} = -M_2^{(L)}$ and we can recognize the action of the compact scalar in 2d with radius $R$.

We are now going to apply the same strategy for theories with $U(1)$ subsystem symmetries, which we will first review in the next session. We will then construct the continuous SymTFT for these cases and study analogous topological and gapless boundary condition on the two side of the SymTFT sandwich respectively.

## 3   Continuous subsystem symmetries

We will study exotic theories with generalizations of continuous dipole symmetries. These are provided by the following conservation law,

$$\partial_0 J_0^I = \partial_i \partial_j \ldots J^K f_K^{ij\ldots,I} \tag{3.1}$$

where the $I$ index on the time component of the current label a representation, $\mathbf{R}_0$, of a continuous or discrete subgroup of the Lorentz group $G \subset SO(d) \subset SO(d,1)$, and similarly the index $K$ on the spatial component, $\mathbf{R}$, so that the current is labelled by a pair of representation $(\mathbf{R}_0, \mathbf{R})$. We can then define the charge,

$$Q(x_\perp^I) = \int_{\Sigma^I} n_I J_0^I \tag{3.2}$$

where $\Sigma^I$ is not generically the full space but a surface labelled by the index $I$, and $n^I$ is its orthogonal vector with norm 1. This implies that the charge itself depends on the coordinates orthogonal to $\Sigma^I$, and it is not generically topological in any direction unless we also have a further conservation equation that sets $\partial_0 J_0^I = 0$.

When building the SymTFT, we will also use this picture to write 1-form gauge fields, some of them will couple to the boundary subsystem currents, in representation $(\mathbf{R}_0, \mathbf{R})$ on the boundary and $(\mathbf{R}_0, \mathbf{R}_r, \mathbf{R})$ in the bulk. We will also use 2-form gauge fields in representation $(\mathbf{R}_{0r}, \mathbf{R}_0, \mathbf{R}_r, \mathbf{R})$. For the representations of the Lorentz's subgroups $\mathbb{Z}_4$ in $2+1d$ and $S_4$ in $3+1d$ we use the notations of [43].

## 3.1 XY-plaquette model

We'll start describing the main properties of the continuous 2+1 dimensional XY-plaquette model; for more information, see [42]. The Lagrangian describing this model is:

$$\mathcal{L}_{xy} = \frac{\mu_0}{2}(\partial_t\phi)^2 - \frac{1}{2\mu}(\partial_x\partial_y\phi)^2 \tag{3.3}$$

where $\phi$ is a real compact scalar of radius $R$: $\phi \sim \phi + 2\pi R(n_x(x) + n_y(y))$. The group of spacetime symmetries is $\mathbf{Z}^4$, representing the groups of discrete $\pi/2$ rotations in the $xy$ plane. We can define an exotic scale invariance for this theory [73]:

$$t \to \lambda^2 t,\ x^i \to \lambda x^i$$
$$\phi \to \phi \tag{3.4}$$

That will be useful to will fix the physical boundary of our SymTFT.

This model also has two $U(1)$ subsystem symmetries. The momentum dipole symmetry, with currents that transform under the $(\mathbf{1}_0, \mathbf{1}_2)$ representation of $\mathbf{Z}^4$ and charges:

$$J_t = \mu_0\partial_t\phi,\ J_{xy} = -\frac{1}{\ },\mu\partial_x\partial_y\phi$$
$$Q_m^x(x) = \oint J_t dy,\ Q_m^y(y) = \oint J_t dx\ , \tag{3.5}$$
$$\oint Q_m^x dx = \oint Q_m^y dy\ ,$$

and the winding dipole symmetry:

$$J_t^{xy} = \frac{1}{2\pi}\partial_x\partial_y\phi,\ J = \frac{1}{2\pi}\partial_t\phi\ ,$$
$$Q_w^x(x) = \oint J_t^{xy} dy,\ Q_w^y(y) = \oint J_t^{xy} dx\ , \tag{3.6}$$
$$\oint Q_w^x dx = \oint Q_w^y dy\ .$$

Where the $\mathbf{Z}^4$ representation of the current is $(\mathbf{1}_2, \mathbf{1}_0)$.

The lagrangian for $\phi$ can be rewritten in terms of the dual field $\phi^{xy}$ [42] as

$$\mathcal{L}_{xy} = \frac{\tilde{\mu}_0}{2}(\partial_t\phi^{xy})^2 - \frac{1}{2\tilde{\mu}}(\partial_x\partial_y\phi^{xy})^2\ , \tag{3.7}$$

with $\tilde{\mu}_0 = \mu/(4\pi^2)$ and $\tilde{\mu} = 4\pi^2\mu_0$. This theory also has a non-invertible duality defect [74].

## 3.2 XYZ-cube model

We will now describe the main properties of the 3+1 dimensional XYZ-cube model [45], described by the Lagrangian:

$$\mathcal{L}_{xyz} = \frac{\mu_0}{2}(\partial_t\phi)^2 - \frac{1}{2\mu}(\partial_x\partial_y\partial_z\phi) \tag{3.8}$$

once again, $\phi$ is a real compact scalar of radius $R$: $\phi \sim \phi + 2\pi R(n_x(x) + n_y(y))$. The group of spacetime symmetries is the orientation preserving subgroup of the cubic group, isomorphic to $S_4$. The scale invariance for this model is defined as:

$$t \to \lambda^3 t,\ x^i \to \lambda x^i$$
$$\phi \to \phi$$

$$(3.9)$$

This model has a momentum quadrupole symmetry:

$$J_0 = \mu_0 \partial_t \phi,\ J_{xyz} = \frac{1}{\mu} \partial_x \partial_y \partial_z \phi$$
$$Q^{ij} = \oint dx^k J_0$$
$$\oint dx^i Q^{ij} = \oint dx^k Q^{ik}$$

$$(3.10)$$

with the current in $(\mathbf{1}, \mathbf{1'})$ representation of $S_4$. There is also a winding quadrupole symmetry:

$$J_0^{xyz} = \frac{1}{2\pi} \partial_x \partial_y \partial_z \phi,\ J = \frac{1}{2\pi} \partial_t \phi$$
$$Q_k^{xyz}(x^i, x^j) = \oint dx^k J_0^{xyz}$$

$$(3.11)$$

with the current in $(\mathbf{1'}, \mathbf{1})$ representation of $S_4$. This Lagrangian is dual to the a model

$$\mathcal{L}_{xyz} = \frac{\hat{\mu}_0}{2} (\partial_t \phi^{xyz})^2 - \frac{1}{2\hat{\mu}} (\partial_x \partial_y \partial_z \phi^{xyz})$$

$$(3.12)$$

where the field $\phi^{xyz}$ is in the $\mathbf{1'}$ representation of $S^4$

### 3.3 $\phi$-model

Two more models can be build in 3+1d [43], the first we describe is the $\phi$ theory:

$$\mathcal{L}_\phi = \frac{\mu_0}{2} (\partial_t \phi)^2 - \frac{1}{4\mu} (\partial_i \partial_j \phi)^2$$

$$(3.13)$$

With scale invariance:

$$t \to \lambda^2 t,\ x^i \to \lambda x^i$$
$$\phi \to \sqrt{\lambda} \phi$$

$$(3.14)$$

This theory has a momentum dipole symmetry:

$$J_0 = \mu_0 \partial_t \phi,\ J_{ij} = -\frac{1}{\mu} \partial_i \partial_j \mu$$
$$Q_{ij}(x^k) = \oint dx^i \oint dx^j J_0$$

$$(3.15)$$

with the current in the $(\mathbf{1}, \mathbf{3}')$ representation of $S_4$, and a winding dipole symmetry:

$$J_0^{ij} = \frac{1}{2\pi}\partial_i\partial_j\phi, \ J = \frac{1}{2\pi}\partial_t\phi$$

$$Q(x^k)[\gamma^{ij}] = \oint_{\gamma^{ij}} J_0^{ik}dx^i + J_0^{jk}dx^j \tag{3.16}$$

where the current is in the $(\mathbf{3}', \mathbf{1})$ representation of $S^4$.

This theory is also dual to the tensor gauge theory with a field $\hat{A}$ in $S_4$ representation $(\mathbf{2}, \mathbf{3}')$:

$$\mathcal{L}_{\hat{A}} = \frac{1}{2\hat{g}_e^2}\left(\partial_t\hat{A}^{ij} - \partial_k A^{k(ij)}\right)^2 - \frac{1}{4\hat{g}_m^2}\left(\partial_i\partial_j\hat{A}^{ij}\right) . \tag{3.17}$$

In the theory the charged operators of the winding dipole can be written explicitly:

$$\hat{W}(x^j, x^k) = e^{i \oint dx^i A^{jk}} . \tag{3.18}$$

## 3.4 $\hat{\phi}$-model

The second model [43] is built using a field $\hat{\phi}^{i(jk)}$ in the $\mathbf{2}$ representation of $S_4$:

$$\mathcal{L}_{\hat{\phi}^{i(jk)}} = \frac{\hat{\mu}_0}{12}\left(\partial_t\hat{\phi}^{i(jk)}\right)^2 - \frac{\hat{\mu}}{4}\left(\partial_k\hat{\phi}^{i(jk)}\right)^2 . \tag{3.19}$$

The scale invariance for this model is much more simple

$$t \to \lambda t, \ x^i \to \lambda x^i$$

$$\phi \to \lambda\phi . \tag{3.20}$$

This model has a momentum tensor symmetry:

$$J_0^{i(jk)} = \hat{\mu}_0\partial_t\hat{\phi}^{i(jk)} , \quad J^{ij} = \hat{\mu}\partial_k\hat{\phi}^{k(ij)}$$

$$J_0^{[ij]k} = \hat{\mu}_0\partial_t\hat{\phi}^{[ij]k} , \quad J^{ij} = \hat{\mu}\partial_k(\hat{\phi}^{[ki]j} + \hat{\phi}^{[kj]i}) \tag{3.21}$$

$$Q^{[ij]}(x^k) = \oint dx^i \oint dx^j J_0^{[ij]k}$$

with current in the $(\mathbf{2}, \mathbf{3}')$ representation of $S_4$. There is also a winding tensor symmetry:

$$J_0^{ij} = \frac{1}{2\pi}\partial_k\hat{\phi}^{k(ij)} , \quad J^{i(jk)}\frac{1}{2\pi}\partial_t\hat{\phi}^{i(jk)}$$

$$Q^k(x^i, x^j) = \oint dx^k J_0^{ij} \tag{3.22}$$

with the current in the $(\mathbf{3}', \mathbf{2})$ representation of $S_4$. This theory is dual to the tensor gauge theory $A$ in $S_4$ representation $(\mathbf{1}, \mathbf{3}')$:

$$\mathcal{L}_A = \frac{1}{2g_e^2}(\partial_t A_{ij} - \partial_i\partial_j A_t) - \frac{1}{2g_m^2}(\partial_i A_{jk} - \partial_j A_{ik})^2 \tag{3.23}$$

where we can write the strip charged under the winding symmetry:

$$W(x_1^k, x_2^k)[\gamma] = \exp\left(i\int_{x_1^k}^{x_2^k} \oint_\gamma A_{ik}dx^i + A_{jk}dx^k\right) . \tag{3.24}$$

# 4 Exotic and foliated SymTFTs, gapless models and dualities

In this section, we introduce the SymTFT for the continuous subsystem symmetries enjoyed by the models described in the previous section. In particular, we will provide two dual descriptions of gapped theories in the bulk, i.e. the exotic and the foliated one.[2] We will then apply the strategy summarized in section 2 by imposing gapped boundary conditions at $M_0$ and scale invariant boundary conditions at $M_L$. In this way we will show how to build dual exotic-foliated gapless models. The gapless physical boundary condition non-linearly realizes the subsystem symmetries providing a gapless theory (or theory of Goldstone bosons) for the spontaneously broken subsystem symmetries. For this reason we will refer to the physical boundary as symmetry breaking boundary. This construction mimics the sandwich in SymTFT, with a key difference that is the foliated nature due to absence of the Lorentz invariance. We call this *Mille-feuille* construction, see figure 1.

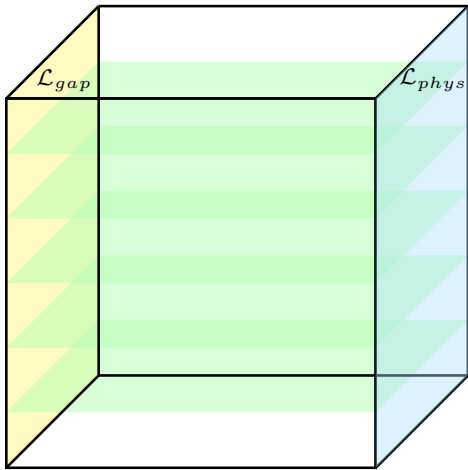

Figure 1: The Mille-feuille. The vertical direction is the foliated one. Some defects of the theory will be topological only on the green layer plane.

The SymTFTs that we will study are gapped theories, but they are not fully topological. More precisely the defects of the SymTFT are partially topological, i.e. they are invariant only under deformations in certain subspaces. This is a consequence of the fact that we are considering theories that are not Lorentz invariant and therefore not every space direction is equivalently treated.

One peculiarity that we will encounter is the linking between semi-topological lines and strip operators. In a 3+1 dimensional manifold, we want to define the $x$-linking

$$\text{Link}_x \left( U(x_u)[\gamma], \tilde{V}(x_1, x_2)[\sigma^x] \right) \tag{4.1}$$

---

[2]Where the notion of duality is the same as in [57].

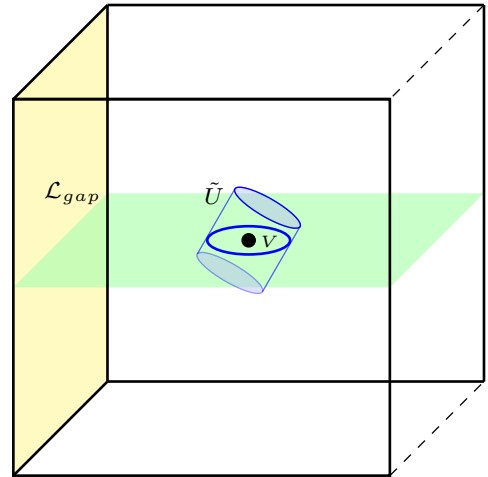

Figure 2: A strip $\tilde{U}$ linking a point operator $V$ on a 2 dimensional submanifold.

between a line $U(x_u)[\gamma]$ living in hyperplane with $x = x_u$ and a strip operator $\tilde{V}(x_1, x_2)[\sigma^x]$ with boundaries on the hyperplanes at $x = x_1$ and $x = x_2$. We define where $\tilde{\gamma}(x')$ as the intersection between $\sigma^x$ and the hyperplane $x = x'$. The definition of such linking is the linking in the $2+1$ hyperplane at $x = x_u$ of the lines $\gamma$ and $\tilde{\gamma}(x_u)$. We show a lower dimensional example in 2+1d in Figure 2, consisting of a strip linking a dot on a 2d submanifold.

## 4.1 XY-plaquette model

In this section, we employ a similar SymTFT construction to the one reviewed in sec. 2. In this case, the SymTFT realizes the continous subsystem symmetries of the (2+1)-dimensional XY-plaquette model. For our purposes, we consider a (3+1)-dimensional bulk $M \times I_r$, where $I$ is the line $r \in [0, L]$, and $M$ is described by coordinates $(x, y, t)$. We denote with $M_0$ the slice at $r = 0$ and with $M_L$ the slice at $r = L$, and $M_0 = -M_L$. A natural choice for the bulk is a topological theory enjoying subsystem symmetries as well, i.e. a foliated BF-type theory with action

$$S = \frac{i}{2\pi} \int_{M \times I} C^x \wedge (dB^x + b) \wedge dx + C^y \wedge (dB^y + b) \wedge dy + b \wedge dc \,, \qquad (4.2)$$

where $b$ and $c$ are a 2-form and a 1-form field defining a usual BF theory, whereas $B^i$ and $C^i$, $i = x, y$, are foliated 1-form fields, namely whose components lie on directions orthogonal to $x^i$. All fields are valued in $\mathbb{R}$. Notice that the bulk terms can be rearranged by integrating by parts, but this has the effect to add terms on the boundary.

Topological edge modes live at the gapped boundary at $r = 0$, ensuring gauge-invariance.

Given the gauge transformations

$$\delta b = d\chi_1 \, ,$$

$$\delta c = d\lambda_0 - \sum_i \lambda^i dx^i \, ,$$

$$\delta B^i = d\chi_0^i - \chi_1 \, ,$$

$$\delta C^i = d\lambda^i \, ,$$

(4.3)

the edge modes have action

$$S_0 = \frac{iR}{2\pi} \int_{M_0} \Phi^x \wedge (d\tilde{B}^x + \tilde{b}) \wedge dx + \Phi^y \wedge (d\tilde{B}^y + \tilde{b}) \wedge dy + \phi \wedge d\tilde{b} \, , \qquad (4.4)$$

where $\tilde{B}^i = B^i|_{z=0}$, $\tilde{b} = b|_{z=0}$ and $\Phi^i$, and $\phi$ are compact, scalar fields of radius $R$ that transform as

$$\delta\phi = -\frac{\lambda_0}{R} \, , \qquad (4.5)$$

$$\delta\Phi^i = -\frac{\lambda^i}{R} \, . \qquad (4.6)$$

The physical or scale-invariant boundary at $z = L$ has action

$$S_L = \frac{1}{4\pi} \int_{M_L} (b + dB^x + dB^y) \wedge *_2 (b + dB^x + dB^y) \wedge dt$$

$$+ g \left(B^x - B^y\right) \wedge *_1 \left(B^x - B^y\right) \wedge dx \wedge dy \, , \qquad (4.7)$$

where $*_2$ and $*_1$ are the Hodge operator on the $x - y$ submanifold and on the time direction only, respectively, and we have inserted a dimensionless coupling $g$ that rescale the term along time.[3] The terms in the previous action are the only possible that preserve scale invariance (3.4) and transform well under $\mathbb{Z}_4$ rotation. Furthermore, notice how the value of the coupling $g$ sits naturally on the physical boundary. This is due to the fact that this coupling parametrizes the dependence of the theory on the physical dimensions of the manifold. On the other hand, $g$ cannot sit on the gapped boundary since the theory on this boundary does not depend on scale redefinition of the coordinates. Note that we have chosen the particular gauge $\chi_1^x = B_x^x$, $\chi_1^y = B_y^y$, since these components do not appear in the action.

The b.c. at the gapped boundary are

$$c = R \left( d\phi + \sum_i \Phi^i dx^i \right) \, ,$$

$$C^i \wedge dx^i = R \, d\Phi^i \wedge dx^i \, , \qquad (4.8)$$

---

[3]The $*_1$ selects the square of the time component, whereas $*_2$ selects the squares of the space components.

while the physical boundary enforces the b.c.

$$c = i *_2 \left( b + \sum_i dB^i \right) \wedge dt \ ,$$

$$C^x \wedge dx = +ig \left( B_t^x - B_t^y \right) dx \wedge dy \wedge dt + id *_2 \left( b + dB^x + dB^y \right) \wedge dt \ ,$$

$$C^y \wedge dy = -ig \left( B_t^x - B_t^y \right) dx \wedge dy \wedge dt + id *_2 \left( b + dB^x + dB^y \right) \wedge dt \ . \tag{4.9}$$

After compactification $L \to 0$, the actions $S_0 + S_L$ in Eqs. (4.7) - (4.4) give the gapless foliated theory dual to the XY-plaquette:

$$S_{\text{gapless}} = \frac{1}{2\pi} \int_{M_0} iR \left[ \Phi^x \wedge (dB^x + b) \wedge dx + \Phi^y \wedge (dB^y + b) \wedge dy + \phi \wedge db \right]$$

$$+ \frac{1}{2} \left( b + dB^x + dB^y \right) \wedge *_2 \left( b + dB^x + dB^y \right) \wedge dt$$

$$+ \frac{1}{2} g \left( B^x - B^y \right) \wedge *_1 \left( B^x - B^y \right) \wedge dx \wedge dy \ . \tag{4.10}$$

By evaluating $S_{\text{gapless}}$ at both b.c. Eqs. (4.9) - (4.8), and integrating out $\Phi^x$, $\Phi^y$, we indeed get the XY-plaquette model. We can also integrate out $\phi$. If the boundary has trivial topology this sets $b = da$. Plugging this back into the action we get a foliated version of a 3d Maxwell theory, with gauge field $a$. This theory is dual to the $XY$-plaquette continuous lagrangian (3.3).

The foliated model living in the bulk is dual [57,74] to the so-called exotic model. In facts, by integrating out the field components $C_r^x$, $C_r^y$, $b_{zx}$, $b_{zy}$, $b_{zt}$ we get the constraints

$$b_{xt} = \partial_t B_x^y - \partial_x B_t^y \ , \quad b_{yt} = \partial_t B_y^x - \partial_y B_t^x$$

$$C_t^x = \partial_x c_t - \partial_t c_x \ , \quad C_t^y = \partial_y c_t - \partial_t c_y \tag{4.11}$$

$$C_y^x + \partial_y c_x = C_x^y + \partial_x c_y \ ,$$

and under the redefinitions

$$A_t = c_t \ , \quad A_r = c_r \ , \quad A_{xy} = C_y^x + \partial_y c_x$$

$$\tilde{A}_t = B_t^x - B_t^y \ , \quad \tilde{A}_r = B_r^x - B_r^y \ , \quad \tilde{A}_{xy} = b_{xy} + \partial_x B_y^x - \partial_y B_x^y \ , \tag{4.12}$$

we obtain the exotic action

$$S = \frac{i}{2\pi} \int d^4x \left[ A_t \left( \partial_r \tilde{A}_{xy} - \partial_x \partial_y \tilde{A}_r \right) + A_r \left( \partial_x \partial_y \tilde{A}_t - \partial_t \tilde{A}_{xy} \right) + A_{xy} \left( \partial_r \tilde{A}_t - \partial_t \tilde{A}_r \right) \right] \ , \tag{4.13}$$

where $d^4x = dr \, dx \, dy \, dt$ is the measure in terms of the coordinates of the space $M \times I_r$.

In the setup that we are considering, for the duality to hold we need the boundaries to be dual as well. The gapped boundary for the exotic model can be obtained from (4.4) by integrating out the scalar fields $\Phi^i$, and the action reads

$$
\begin{aligned}
S_0 &= \frac{iR}{2\pi} \int d^3x \, \phi \left[ \partial_t \left( b_{xy} + \partial_x B_y^x - \partial_y B_x^y \right) - \partial_x \partial_y \left( B_t^x - B_t^y \right) \right] \\
&= \frac{iR}{2\pi} \int d^3x \, \phi \left( \partial_t \tilde{A}_{xy} - \partial_x \partial_y \tilde{A}_t \right) \, ,
\end{aligned}
\tag{4.14}
$$

where $d^3x = dx \, dy \, dt$ is the measure in terms of the coordinates of the space $M_0$. The bulk-boundary action is invariant under the gauge-transformations

$$
\delta A_t = \partial_t \lambda \, , \quad \delta \tilde{A}_t = \partial_t \tilde{\lambda}
$$

$$
\delta A_r = \partial_r \lambda \, , \quad \delta \tilde{A}_r = \partial_r \tilde{\lambda}
$$

$$
\delta A_{xy} = \partial_x \partial_y \lambda \, , \quad \delta \tilde{A}_{xy} = \partial_x \partial_y \tilde{\lambda} \, .
\tag{4.15}
$$

In addition the boundary variational problem gives

$$
A_t = -R \left( \partial_t \phi \right) \, ,
$$

$$
A_{xy} = -R \left( \partial_x \partial_y \phi \right) \, .
\tag{4.16}
$$

Finally, the scale invariant boundary term dual, obtained by applying the redefinition (4.12) to (4.7), is given by

$$
S_L = \frac{1}{4\pi} \int_{M_L} d^3x \left[ g(\tilde{A}_t)^2 - (\tilde{A}_{xy})^2 \right] \, ,
\tag{4.17}
$$

where $d^3x = dx \, dy \, dt$ is the measure on $M_L$. This boundary action leads to

$$
\tilde{A}_t = \frac{i}{g} A_{xy} \, ,
$$

$$
\tilde{A}_{xy} = -i A_t \, .
\tag{4.18}
$$

There are more gauge transformations in the foliated case than in the exotic theory, compare (4.3) with (4.15). Some of them will become redundant by the identification (4.12), like in [57].

Compactifying the interval by sending $L \to 0$, both b.c. (4.16)-(4.18) must be valid and we obtain

$$
S_{xy} = \frac{1}{4\pi} \int_M d^3x \left[ R^2 (\partial_t \phi)^2 - \frac{R^2}{g} (\partial_x \partial_y \phi)^2 \right] \, .
\tag{4.19}
$$

Finally, by identifying the coupling as

$$
R^2 = \mu_0 \, ,
$$

$$
\frac{R^2}{g} = \frac{1}{\mu} \, ,
\tag{4.20}
$$

we get the action of the XY-plaquette model in (3.3).

## Spectrum of gauge-invariant operators

We construct the gauge-invariant operators in the bulk of both models, the foliated theory and the exotic theory. Let us start with the foliated model. From the action in (4.2), the equations of motion read

$$(dB^i + b) \wedge dx^i = 0 \ ,$$

$$db = 0 \ ,$$

$$dC^i \wedge dx^i = 0 \ ,$$

$$\sum_i dC^i \wedge dx^i + dc = 0 \ . \tag{4.21}$$

There are two types of partially topological lines

$$V_\alpha(x,y)[\gamma] = \exp\left(i\alpha \oint_\gamma c\right) \ ,$$

$$U_\beta(x,y)[\gamma] = \exp\left(i\beta \oint_\gamma B^x - B^y\right) \ , \tag{4.22}$$

where $\alpha$, $\beta \in \mathbb{R}$, and $\gamma$ is a closed curve inside a 2-dimensional submanifold defined at fixed $(x,y)$. Inside this submanifold, these lines can be topologically deformed, but not on the whole 4-dimensional volume.

Furthermore, we can define two surface operators. The first one is

$$\tilde{U}_{\tilde\beta}[\Sigma] = \exp\left(i\tilde\beta \oint_\Sigma b\right) \ , \tag{4.23}$$

where $\Sigma$ is a closed surface in the bulk. Notice that for the foliated nature of the theory in the bulk, we can open the surface $\Sigma$ and restore gauge-invariance by adding the foliated operators. In other words, we can define the surface operator on a strip $\sigma^i$ as

$$\tilde{U}_{\tilde\beta}^i(x_1^i, x_2^i)[\sigma^i] = \exp\left\{i\tilde\beta \int_{x_1^i}^{x_2^i}\left(\oint_{\gamma_i(x^i)} b + dB^i\right) dx^i\right\} \ . \tag{4.24}$$

We can deform this strip, but we cannot move the end manifolds at $x_1^i$ and $x_2^i$. The last operator is defined on a strip as well and it reads

$$\tilde{V}_{\tilde\alpha}^i(x_1^i, x_2^i)[\sigma^i] = \exp\left\{i\tilde\alpha \int_{x_1^i}^{x_2^i}\left(\oint_{\gamma_i(x^i)} C^i \wedge dx^i + d\left(c_i dx^i\right)\right) dx^i\right\} \ , \tag{4.25}$$

where in $d(c_i dx^i)$ the index is not summed over. From the e.o.m. with the defect insertions, we can write their braiding

$$\langle V_\alpha[\gamma], \tilde{U}_{\tilde\beta}[\sigma^i]\rangle = \exp\left(2\pi i\alpha\tilde\beta \operatorname{Link}_i(\gamma, \gamma^i)\right) \ ,$$

$$\langle U_\beta[\gamma], \tilde{V}_{\tilde{\alpha}}[\sigma^i] \rangle = \exp\left(2\pi i \beta \tilde{\alpha} \operatorname{Link}_i(\gamma, \gamma^i)\right) . \tag{4.26}$$

Turning to the exotic theory, the e.o.m. reads

$$\partial_t A_r - \partial_r A_t = 0 ,$$
$$\partial_x \partial_y A_t - \partial_t A_{xy} = 0 ,$$
$$\partial_x \partial_y A_r - \partial_r A_{xy} = 0 ,$$
$$\partial_t \tilde{A}_r - \partial_r \tilde{A}_t = 0 ,$$
$$\partial_x \partial_y \tilde{A}_t - \partial_t \tilde{A}_{xy} = 0 ,$$
$$\partial_x \partial_y \tilde{A}_r - \partial_r \tilde{A}_{xy} = 0 . \tag{4.27}$$

The set of operators for the exotic theory consists of the lines

$$V_\alpha(x, y)[\gamma] = \exp\left(i\alpha \oint_\gamma A_t dt + A_r dz\right) ,$$

$$U_\beta(x, y)[\gamma] = \exp\left(i\beta \oint_\gamma \tilde{A}_t dt + \tilde{A}_r dz\right) , \tag{4.28}$$

and the operators defined on a strip

$$\tilde{V}_{\tilde{\alpha}}^i(x_1^i, x_2^i)[\sigma^i] = \exp\left\{i\tilde{\alpha} \int_{x_1^i}^{x_2^i} \left(\oint_{\gamma_i(x^i)} A_{xy} dx^{(j)} + \partial_i A_r dx + \partial_i A_t dt\right) dx^i\right\} , \quad j \neq i ,$$

$$\tilde{U}_{\tilde{\beta}}^i(x_1^i, x_2^i)[\sigma^i] = \exp\left\{i\tilde{\beta} \int_{x_1^i}^{x_2^i} \left(\oint_{\gamma_i(x^i)} \tilde{A}_{xy} dx^{(j)} + \partial_i \tilde{A}_r dx + \partial_i \tilde{A}_t dt\right) dx^i\right\} , \quad j \neq i . \tag{4.29}$$

These operators correspond to the set of lines and strips of the foliated theory by the identification (4.12).

## Boundary operators

The operators on the XY-plaquette model obtained after compactification can be identified with the set of operators defined in the SymTFT, given that the gapped boundary forces flux quantization as

$$\int_{\Sigma \subset M_0} \tilde{b} \in \frac{2\pi}{R} \mathbb{Z} ,$$

$$\oint_{\gamma \subset M_0} \tilde{B}^i = \int_{x_1^i}^{x_2^i} \oint_{\gamma \subset M_0} \tilde{B}^i \in \frac{2\pi}{R} \mathbb{Z} . \tag{4.30}$$

The set of lines $U_\alpha$ and $V_\beta$ in Eq. (4.22) trivializes if $\alpha = p/R$ and $\beta = q^i/R$, with $p, q^i \in \mathbb{Z}$, thereby ending on the boundary. The non-trivial lines projected on the boundary are identified as

$$\alpha \sim \alpha + R , \quad \beta \sim \beta + R , \tag{4.31}$$

and correspond to the operators generating the subsystem $U(1) \times U(1)$ symmetry of the XY-plaquette.

On the other hand, when the strip operator $\tilde{V}$ (4.25) ends on the boundary, it corresponds to the operators charged under the momentum dipole $U(1)$ symmetry of the boundary

$$\exp\left\{i\int_{x_1^i}^{x_2^i}\left(\oint_{\gamma_i(x^i)}C^i\wedge dx^i + d\left(c_i dx^i\right)\right)dx^i\right\} = \exp\left\{i\int_{x_1^i}^{x_2^i}Q_m^i(x^i)\right\}, \tag{4.32}$$

while $\tilde{U}$ (4.24) corresponds to operators charged under the winding $U(1)$

$$\exp\left\{i\int_{x_1^i}^{x_2^i}\left(\oint_{\gamma_i(x^i)}b + dB^i\right)dx^i\right\} = \exp\left\{i\int_{x_1^i}^{x_2^i}Q_m^i(x^i)\right\}, \tag{4.33}$$

both carrying integer charges [42].

### 4.1.1 Exotic-foliated interfaces

We can interpret the condition (4.12) as an interface between the gapped bulk exotic (on the left of the interface) and foliated (on the right of the interface) SymTFT, see 3, where

$$S_I = \frac{i}{2\pi}\int_{M_I}\left[\Phi^x\partial_y\left(\tilde{A}^t - B_t^x - B_t^y\right) - \Phi^y\partial_x\left(\tilde{A}^t - B_t^x - B_t^y\right)\right.$$
$$\left. + R\,\phi\,\partial_x\partial_y\left(\tilde{A}^t - B_t^x - B_t^y\right)\right]dx\,dy\,dt \tag{4.34}$$

where $\Phi^{(x,y)}$ are fields that live on the interface and their name is chosen because they will correspond indeed to the fields in (4.8) with the same name. When we fuse the interface with the gapped boundary for the exotic theory, i.e. the second line in (4.14). The $\tilde{A}^t, \tilde{A}^{xy}$ field become related to the boundary fields, $\tilde{b}$, by

$$\tilde{A}^{xy}|_{M_0=-M_I} = \tilde{b}_{xy} + \partial_x B_y^x - \partial_y B_x^y,$$
$$\partial_x\tilde{A}^t|_{M_0=-M_I} = \tilde{b}_{xt} + \partial_x B_t^x - \partial_t B_x^y, \tag{4.35}$$
$$\partial_y\tilde{A}^t|_{M_0=-M_I} = \tilde{b}_{yt} + \partial_y B_t^y - \partial_t B_y^x$$

These equations are compatible with the condition (4.12) and with the constraints (4.11). Substituting this into (4.34) and the second line of (4.14) we obtain (4.4). The interfaces for the cases in the following subsection can be similarly written, we plan to come back to this in the future.

### 4.1.2 Condensation defect

The theory in the bulk enjoys a symmetry that exchanges the non-local operators, i.e. T-duality. We can construct a condensation defect in the bulk that generates this symmetry,

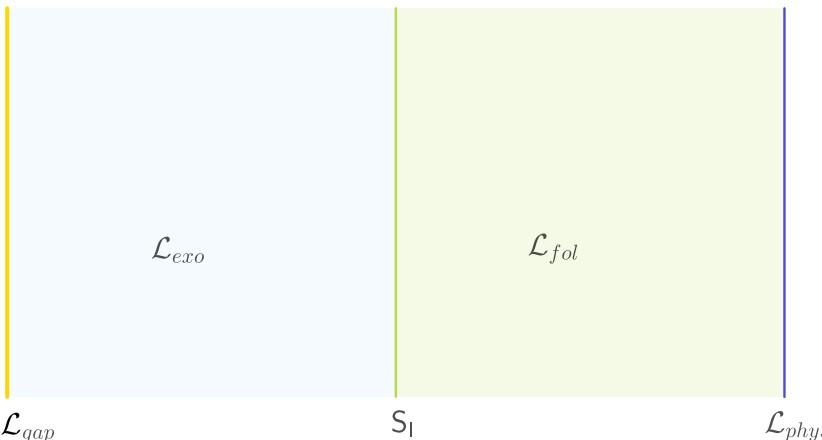

Figure 3: The interface $S_I$ that implements the duality map between foliated and exotic theories.

following the idea in [65]. For simplicity, we introduce it in the exotic model. The condensation defect is supported on a 3-dimensional subspace $\Gamma$, which can extend in either the $(z,t)$ plane. Without loss of generality, let us place it at a position $z = \bar{z}$. It can always be rotated on the $(z,t)$ plane. Due to the subsystem symmetry, this defects can be opened with a boundary only on $t$. We can write its action as

$$S^{\mathcal{C}^T}[\Gamma] = \int_{z=\bar{z}} \left[ \psi \left( \partial_t \tilde{E}^{xy} - \partial_x \partial_y \tilde{E}^t \right) + \tilde{E}^t \left( A^{xy} - \tilde{A}_{xy} \right) + \tilde{E}^{xy} \left( A^t - \tilde{A}_t \right) \right]$$
$$+ \int_{\partial\Gamma} \left[ \tilde{\sigma} \left( A^{xy} - \tilde{A}_{xy} \right) - \psi \left( \tilde{E}^{xy} + \partial_x \partial_y \tilde{\sigma} \right) \right] , \tag{4.36}$$

where $\psi$ is a scalar, $\tilde{E}^{xy}$, $\tilde{E}^t$ are gauge fields and $\tilde{\sigma}$ is an edge mode that ensures gauge invariance, together with the e.o.m. of the bulk fields, given that

$$\delta\psi = \lambda - \tilde{\lambda} ,$$
$$\delta\tilde{E}^{xy} = \partial_x \partial_y \tilde{\Lambda} ,$$
$$\delta\tilde{E}^t = \partial_t \tilde{\Lambda} ,$$
$$\delta\tilde{\sigma} = -\tilde{\Lambda} . \tag{4.37}$$

The gauge fields ensure that the non-local operators in (4.28) - (4.29) are exchanged. By fusing the condensation defect with its conjugated, we obtain the non-invertible duality defect of [74].

## 4.2 XYZ-cube model

The theories we are now going to discuss live on $3+1$ dimensional manifold $M_4$. The SymTFTs will therefore live on $M_4 \times I$ with boundaries $M_0$ and $M_L$, which are two copies of $M_4$. We will call the boundary coordinates $t, x, y, z$ and the interval (or bulk) coordinate $r$, we will also use $i, j, k$ to indicate the boundary space coordinates.

### 4.2.1 Exotic SymTFT

In strong analogy with the $XY$-cube studied in the previous section, we propose a SymTFT to describe $U(1)$ quadrupole symmetries in dimension $3 + 1$:

$$\mathcal{L}_{sym_{xyz}} = \frac{i}{2\pi} \left[ \tilde{A}_t(\partial_r A_{xyz} - \partial_x \partial_y \partial_z A_r) + \tilde{A}_r(\partial_x \partial_y \partial_z A_t - \partial_t A_{xyz}) + \tilde{A}_{xyz}(\partial_t A_r - \partial_r A_t) \right]$$
$$(4.38)$$

where both $A$ and $\tilde{A}$ are $\mathbb{R}$-gauge field with gauge transformations:

$$\delta A_t = \partial_t \lambda, \; \delta \tilde{A}_t = \partial_t \tilde{\lambda}$$
$$\delta A_r = \partial_r \lambda, \; \delta \tilde{A}_r = \partial_r \tilde{\lambda} \quad\quad (4.39)$$
$$\delta A_{xyz} = \partial_x \partial_y \partial_z \lambda, \; \delta \tilde{A}_{xyz} = \partial_x \partial_y \partial_z \tilde{\lambda}$$

and the equations of motion are:

$$\partial_t A_r - \partial_r A_t = 0, \; \partial_t \tilde{A}_r - \partial_r \tilde{A}_t = 0$$
$$\partial_t A_{xyz} - \partial_x \partial_y \partial_z A_t = 0, \; \partial_t \tilde{A}_{xyz} - \partial_x \partial_y \partial_z \tilde{A}_t = 0 \quad\quad (4.40)$$
$$\partial_r A_{xyz} - \partial_x \partial_y \partial_z A_r = 0, \; \partial_r \tilde{A}_{xyz} - \partial_x \partial_y \partial_z \tilde{A}_r = 0 \, .$$

### 4.2.2 Foliated SymTFT

We can build a foliated SymTFT with single and double foliations, describing the same model:

$$\mathcal{L}_{sym_{xyz}} = \frac{i}{2\pi} \left( b \wedge dc + B^i \wedge dC^i \wedge dx^i + \mathbf{B}^{ij} \wedge \mathbf{C}^{ij} \wedge dx^i \wedge dx^j \right.$$
$$\left. + b \wedge C^i \wedge dx^i + B^i \wedge \mathbf{C}^{ij} \wedge dx^i \wedge dx^j \right) \quad\quad (4.41)$$

where b is a 3-form $\mathbb{R}$ gauge field, the $B^i$ are 2-form $\mathbb{R}$ gauge fields and all the others are 1-form $\mathbb{R}$ gauge fields. The gauge transformations are:

$$\delta b = d\chi_2$$
$$\delta B^i = d\chi_1^i - \chi^2$$
$$\delta \mathbf{B}^{ij} = d\chi_0^{ij} - \chi_1^i - \chi_1^j$$
$$\delta c = d\lambda - \sum_i \lambda^i \wedge dx^i \qquad (4.42)$$
$$\delta C^i = d\lambda^i - \sum_j \lambda^{ij} \wedge dx^j j$$
$$\delta \mathbf{C}^{ij} = d\lambda^{ij} \ ,$$

and the equations of motion are:

$$db = 0$$
$$dc + C^i \wedge dx^i = 0$$
$$(dB^i + b) \wedge dx^i = 0$$
$$dC^i \wedge dx^i + \sum_j \mathbf{C}^{ij} \wedge dx^i \wedge dx^j = 0 \qquad (4.43)$$
$$(d\mathbf{B}^{ij} + B^i - B^j) \wedge dx^i \wedge dx^j = 0$$
$$d\mathbf{C}^{ij} \wedge dx^i \wedge dx^j = 0$$

### 4.2.3 Gauge invariant operators

We will now describe the gauge invariant operators present in this theory. Given the nature of this theory, such operators will be semi-topological in nature. They will give us two important results: by showing that the two theories have the same gauge invariant operators, we can show that these two theories are indeed dual [57]. Furthermore, we will be able to use such operators in the SymTFT picture to describe the symmetry operators (3.15), (3.16) and the charged operators.

We can check for gauge invariance using the gauge transformations in the previous section and the semi-topological nature using the equation of motion of both theories. We will provide the expression of the operators for the dual theories together. The duality between the line is a consequence of the identification,

$$A_t = \mathbf{B}_t^{xy} + \mathbf{B}_t^{yz} + \mathbf{B}_t^{xz}, \ A_r = \mathbf{B}_r^{xy} + \mathbf{B}_r^{yz} + \mathbf{B}_r^{xz},$$
$$A_{xyz} = b_{xyz} + \partial_x B_{yz}^x - \partial_z B_{xy}^r + \partial_y B_{zx}^y + \partial_x \partial_y \mathbf{B}_z^{xy} - \partial_x \partial_z \mathbf{B}_y^{xz} + \partial_y \partial_z \mathbf{B}_x^{yz} \qquad (4.44)$$
$$\tilde{A}_t = c_t, \ \tilde{A}_r = c_r, \ \tilde{A}_{xyz} = \mathbf{C}_z^{xy} + \partial_x C_z^y + \partial_x \partial_y c_z$$

We start with the line operators.

$$V_\alpha(x,y,z)[\gamma] = \exp\left(i\alpha \oint \tilde{A}_t dt + \tilde{A}_r dr\right)$$
$$= \exp\left(i\alpha \oint_\gamma c\right)$$
$$U_\alpha(x,y,z)[\gamma] = \exp\left(i\alpha \oint A_t dt + A_r dr\right)$$
$$= \exp\left(i\alpha \oint_\gamma B^{xy} - B^{yz} + B^{zx}\right)$$

(4.45)

We then have two copies of "bar" operators on the solid bar $\rho^{ij}$:

$$\tilde{V}_\alpha(x_1^i, x_2^i, x_1^j, x_2^j)[\rho^{ij}] = \exp\left(i\alpha \int_{x_1^i}^{x_2^i} \int_{x_1^j}^{x_2^j} \oint_{\gamma(x^i,x^j)} \partial_i \partial_j \tilde{A}_t dt + \tilde{A}_{xyz} dx^k\right)$$
$$= \exp\left(i \int_{x_1^i}^{x_2^i} \int_{x_1^j}^{x_2^j} \oint C^{ij} \wedge dx^i \wedge dx^j + d(C^i \wedge dx^i + d(c_i dx^i))_{ij} dx^i \wedge dx^j)\right)$$

$$\tilde{U}_\alpha(x_1^i, x_2^i, x_1^j, x_2^j)[\rho^{ij}] = \exp\left(i\alpha \int_{x_1^i}^{x_2^i} \int_{x_1^j}^{x_2^j} \oint_{\gamma(x^i,x^j)} \partial_i \partial_j A_t dt + A_{xyz} dx^k\right)$$
$$= \exp\left(i\alpha \int_{x_1^i}^{x_2^i} \int_{x_1^j}^{x_2^j} \oint b + \sum_{ijk} d(B_{jk}^i dx^j \wedge dx^k) + \sum_{ijk} d(d(\mathbf{B}_k^{ij} dx^k)_{jk} dx^j \wedge dx^k)\right),$$

(4.46)

where in $d(d(\mathbf{B}_k^{ij} dx^k)_{jk}$ the indices are not summed over.

We can then compute the linking of a the operators:

$$\langle \tilde{V}_\alpha(x_1^i, x_2^i, x_1^j, x_2^j)[\rho^{ij}], U_\beta(x,y,z)[\gamma]\rangle = e^{2\pi i\alpha\beta \mathrm{Link}_{ij}(\gamma, \rho^{ij})}$$
$$\langle \tilde{U}_\alpha(x_1^i, x_2^i, x_1^j, x_2^j)[\rho^{ij}], V_\beta(x,y,z)[\gamma]\rangle = e^{2\pi i\alpha\beta \mathrm{Link}_{ij}(\gamma, \rho^{ij})}$$

(4.47)

### 4.2.4 Exotic edge mode

We can now make the theory with boundary gauge invariant with a gapped edge mode:

$$\mathcal{L}_{\bar{\Sigma}} = \frac{iR}{2\pi} \phi(\partial_x \partial_y \partial_z A_t - \partial_t A_{xyz})$$
$$\delta\phi = -\frac{\lambda}{R}$$

(4.48)

where $\phi$ is a compact real scalar subject to the discrete gauge symmetry:

$$\phi \sim \phi + 2\pi\left(n(x,y) + n(y,z) + n(z,x)\right) .$$

(4.49)

We can now add a scale invariant edge mode on $M_L$:

$$\mathcal{L}_\Sigma = \frac{1}{4\pi}(\mu_0(A_t)^2 - \frac{1}{\mu}(A_{xyz})^2) .$$

(4.50)

We can then perform the slab compactification and get:

$$\mathcal{L}_{xyz} = \frac{1}{2\pi} \left( \frac{1}{2\mu}(A_t)^2 - \frac{\mu_0}{2}(A_{xyz})^2 - iR(\phi(\partial_x \partial_y \partial_z A_t - \partial_t A_{xyz})) \right) \tag{4.51}$$

which, after integrating out $A_t$ and $A_{xyz}$ becomes the XYZ-cube model (3.8).

### 4.2.5  Foliated edge mode

The corresponding foliated gapped edge mode is:

$$\mathcal{L}_{\bar{\Sigma}} = \frac{iR}{2\pi}(\phi \wedge db + \Phi^i \wedge dB \wedge dx^i + \mathbf{\Phi}^{ij} \wedge d\mathbf{B}^{ij} \wedge dx^i \wedge dx^j$$
$$+ \Phi^i \wedge b \wedge dx^i + \mathbf{\Phi}^{ij} \wedge B^i \wedge dx^i \wedge dx^j) \tag{4.52}$$
$$\delta\phi = -\frac{\lambda}{R}, \ \delta\Phi^i = -\frac{\lambda^i}{R}, \ \delta\mathbf{\Phi}^{ij} = -\frac{\lambda^{ij}}{R},$$

where $\phi$, $\Phi^i$ and $\mathbf{\Phi}^{ij}$ are compact scalars of radius $R$. Notice that, because of the foliation, the fields $\Phi^i$ and $\phi^{ij}$ can be discontinuous along the foliated directions. This correspond to the boundary conditions:

$$c = d\phi + \Phi^i \wedge dx^i$$
$$C^i \wedge dx^i = d\Phi^i \wedge dx^i + \mathbf{\Phi}^{ij} \wedge dx^i \wedge dx^j$$
$$\mathbf{C}^{ij} = d\mathbf{\Phi}^{ij}$$
$$db = 0$$
$$(dB^i + b) \wedge dx^i = 0$$
$$(d\mathbf{B}^{ij} + B^i - B^j) \wedge dx^i \wedge dx^j = 0 \tag{4.53}$$
$$\oint b \in \frac{2\pi}{R}\mathbb{Z}$$
$$\int \oint B^i \wedge dx^i \in \frac{2\pi}{R}\mathbb{Z}$$
$$\int \int \oint \mathbf{B}^{ij} \wedge dx^i \wedge dx^j \in \frac{2\pi}{R}\mathbb{Z} \ .$$

We can then write the scale invariant edge mode:

$$\mathcal{L}_{\Sigma} = \frac{1}{4\pi} \left( (b + \sum_{ijk} d(B^i_{jk}dx^j \wedge dx^k) + \sum_{ijk} d(d(\mathbf{B}^{ij}_k dx^k)_{jk}dx^j \wedge dx^k) \right) *_3 \wedge (b + ...) \wedge dt$$
$$+ (\mathbf{B}^{xy} - \mathbf{B}^{yz} + \mathbf{B}^{zx}) *_1 \wedge (\mathbf{B}^{xy} - \mathbf{B}^{yz} + \mathbf{B}^{zx}) \wedge dx \wedge dy \wedge dz \ . \tag{4.54}$$

Writing together the two expressions (4.52) and (4.54) we can write the dual version of the XYZ cube (3.8). We can also integrate out $\phi$, setting $b = da_2$ when the topology of $M_4$ is trivial, and we obtain a foliated version of the 2-form Maxwell gauge theory in 4d, which is dual to the XYZ-cube Lagrangian (3.8).

### 4.2.6 Boundary operators

We are now interested in the parallel projection of the operators on the boundary. We can see from the linking (4.47) that the lines are charged under the bar operators. More specifically, they can end on the boundary if their parameter $\alpha$ is quantized as $2\pi n R$ and $2\pi/R$ respectively, becoming the charged objects $\phi$ and $\phi^{xyz}$. The bar operators, inserted along the space direction, become the charges in eq (3.10) and (3.11):

$$
\begin{aligned}
\tilde{V}_\alpha(x_1^i, x_2^i, x_1^j, x_2^j) &= \exp\left(i\alpha \int_{x_1^i}^{x_2^i} \int_{x_1^j}^{x_2^j} Q^{ij}\right) \\
\tilde{U}_\alpha(x_1^i, x_2^i, x_1^j, x_2^j) &= \exp\left(i\alpha \int_{x_1^i}^{x_2^i} \int_{x_1^j}^{x_2^j} Q_k^{xyz}\right)
\end{aligned}
\tag{4.55}
$$

## 4.3 $\phi$-model

### 4.3.1 Exotic SymTFT

In this section we propose a new exotic SymTFT describing $U(1)$ dipole symmetries in dimension 3+1:

$$
\mathcal{L}_{sym-\phi} = \frac{i}{2\pi}(F_r^{ij}(\partial_t A_{ij} - \partial_i\partial_j A_t) + F_t^{ij}(\partial_i\partial_j A_r - \partial_r A_{ij}) + F(\partial_t A_r - \partial_r A_t) + F_{rt}^{[ij]k}(\partial_i A_{jk} - \partial_j A_{ik}))
\tag{4.56}
$$

where we follow the index notation of [42], it is clear from the context that in this case the upper indices do not indicate any foliation. The exotic 1-form $\mathbb{R}$-gauge field $A$ is in $S_4$ representation $(\mathbf{1}, \mathbf{1}, \mathbf{2})$ with gauge transformations:

$$
\delta A_t = \partial_t\lambda, \ \delta A_r = \partial_r\lambda, \ \delta A_{ij} = \partial_i\partial_i\lambda
\tag{4.57}
$$

and the exotic 2-form $\mathbb{R}$-gauge field $F$ is in $S_4$ representation $(\mathbf{2}, \mathbf{3'}, \mathbf{3'}, \mathbf{1})$. The gauge transformation read

$$
\begin{aligned}
\delta F_{tr}^{[ij]k} &= \partial_t\chi_r^{[ij]k} - \partial_r\chi_t^{[ij]k} \\
\delta F_t^{ij} &= \partial_t\chi^{ij} - \partial_k\chi_t^{k(ij)} \\
\delta F_r^{ij} &= \partial_r\chi^{ij} - \partial_k\chi_r^{k(ij)} \\
\delta F &= \frac{1}{2}\partial_i\partial_j\chi^{ij}
\end{aligned}
\tag{4.58}
$$

where $\chi$ is an exotic gauge parameter in $S_4$ representation $(\mathbf{2}, \mathbf{2}, \mathbf{3'})$. We can also write the equation of motion, they will be useful to study the semi-topological nature of the gauge

invariant operators:

$$\partial_t A_{ij} - \partial_i \partial_j A_t = 0$$
$$\partial_r A_{ij} - \partial_i \partial_j A_r = 0$$
$$\partial_t A_r - \partial_r A_t = 0$$
$$\partial_i A_{jk} - \partial_j A_{ik} = 0 \tag{4.59}$$
$$\partial_i \partial_j F_t^{ij} - \partial_t F = 0$$
$$\partial_i \partial_j F_r^{ij} - \partial_r F = 0$$
$$\partial_r F_t^{ij} - \partial_t F_r^{ij} + \partial_k F^{k(ij)} = 0.$$

### 4.3.2 Foliated SymTFT

We can describe the same symmetries using a foliated BF theory:

$$\mathcal{L}_{sym-\phi} = \frac{1}{2\pi}(c \wedge db + C^i \wedge dB^i \wedge dx^i + b \wedge C^i \wedge dx^i), \tag{4.60}$$

where $b$ is a 3-form $\mathbb{R}$ gauge field, $B^i$ is a 2-form $\mathbb{R}$ gauge field, $c$ is a 1-form $\mathbb{R}$ gauge field and $C^i$ is a 1-form foliated $\mathbb{R}$ gauge field.

The related gauge transformations are given by

$$\delta b = d\chi_2$$
$$\delta c = d\lambda_0 - \sum_i \lambda^i dx^i$$
$$\delta B^i = d\chi_1 - \mu_2 \tag{4.61}$$
$$\delta C^i = d\lambda^i,$$

and the equation of motion for this theory read

$$db = 0$$
$$dB^i \wedge dx^i - b \wedge dx^i = 0$$
$$dc + \sum_i C^i \wedge dx^i = 0 \tag{4.62}$$
$$dC^i \wedge dx^i = 0$$

### 4.3.3 Gauge invariant operators

We will now study the gauge invariant and semi-topological defects of the two theories. We will present the defects of the dual theories together, where their identification is a consequence of

$$A_{ij} = C_j^i + \partial_j c_i, \ A_t = c_t, \ A_r = c_r$$
$$F_t^{ij} = B_{tj}^i - B_{ti}^j, \ F = b_{xyz} + \partial_x B_{yz}^x - \partial_y B_{xz}^y + \partial_z B_{xy}^r \tag{4.63}$$
$$F^{[ij]k} = B_{rt}^i - B_{rt}^j$$

It can be seen that, under these transformations, the equation of motions are matched and, after integrating out some constraints, the two Lagrangians transform into each other.

Both theories have a gauge invariant line operator topological in the $th$ plane:

$$V_\alpha(x,y,z)[\gamma] = \exp\left(i\alpha \int_\gamma A_t dt + A_r dr\right) = \exp\left(i\alpha \int_\gamma c\right) \tag{4.64}$$

We can then write two corresponding strip operators:

$$\tilde{V}_\alpha^{jk}(x_1^i, x_2^i)[\sigma^i] = \exp\left(i\alpha \int_{x_1^i}^{x_2^i} dx^i \oint_{\gamma_i(x^i)} A_{ik} dx^k + A_{ij} dx^j + \partial_i A_r dr + \partial_i A_t dt\right)$$
$$= \exp\left(i\alpha \int_{x_1^i}^{x_2^i} dx^i \oint_{\gamma_i(x^i)} C^i \wedge dx^i + d(c_i dx^i)\right) \tag{4.65}$$

where in $\tilde{V}^{jk}$ the indices denote the $(x^j, x^k, r, t)$ hyperplane where the operator is supported.

Using the remaining fields we can build a surface operator topological in the $rtx^k$ volume:

$$U_\alpha^k(x^i, x^j)[\Gamma] = \exp\left(i\alpha \oint_\Gamma F_r^{ij} dr dx^k + F_t^{ij} dt dx^k + F^{k(ij)} dr dt\right)$$
$$= \exp\left(i\alpha \oint_\Gamma B^i - B^j\right) \tag{4.66}$$

where in $\tilde{U}^k$ the index denotes the $(x^k, r, t)$ hyperplane where the operator is supported, and two dual "slab" operators on the slab $\Sigma^k$:

$$\tilde{U}_\alpha^{ij}(x_1^k, x_2^k)[\Sigma^k] = \exp\left(i\alpha \int_{x_2^k}^{x_1^k} \oint_{\Gamma(x^k)} F dx^i dx^j + \partial_k F_t^{jk} dt dx^i + (\partial_j F_t^{ij} + \partial_k F_t^{ik}) dt dx^j\right)$$
$$= \exp\left(i\alpha \int \oint_{\Gamma(x^k)} b + \sum_{ijk} d(B_{jk}^i dx^j \wedge dx^k)\right) \tag{4.67}$$

We also want to describe how the operators we described act on each other via linking. We can study the effect of the insertion of these defects on the equations of motion and find that their braiding is:

$$\langle V_\alpha(x,y,z)[\gamma], \tilde{U}_\beta^{ij}(x_1^k, x_2^k)[\Sigma^k]\rangle = e^{2\pi i \alpha\beta \mathrm{Link}_k(\gamma, \Sigma^k)}$$
$$\langle U_\alpha^k(x^i, x^j)[\Gamma], \tilde{V}_\beta^{jk}(x_1^i, x_2^i)[\sigma^i]\rangle = e^{2\pi i \alpha\beta \mathrm{Link}_i(\Gamma, \sigma^i)} \tag{4.68}$$

### 4.3.4 Exotic edge modes

Under gauge transformation for the theory (4.56) we get a boundary term of the form:

$$\delta S = \frac{i}{2\pi} \int_{\bar{\Sigma}} \chi^{ij}(\partial_t A_{ij} - \partial_i \partial_j A_t) + \chi_t^{[ij]k}(\partial_i A_{jk} - \partial_j A_{ik}) \tag{4.69}$$

The gapped edge mode on the boundary will therefore have an exotic 1-form $U(1)$-gauge field $\tilde{A}$ in representation $(\mathbf{2}, \mathbf{3}')$:

$$
\begin{aligned}
\mathcal{L}_{\bar{\Sigma}} &= \frac{i}{2\pi R}\tilde{A}_{ij}(\partial_t A_{ij} - \partial_i\partial_j A_t) + \tilde{A}_t^{[ij]k}(\partial_i A_{jk} - \partial_j A_{ik}) \\
\delta\tilde{A}_t^{[ij]k} &= -\chi_t^{[ij]k}R,\ \delta\tilde{A}_{ij} = -\chi^{ij}R
\end{aligned}
\tag{4.70}
$$

This is the Lagrangian of the X-cube model [41], the only difference with the model being the absence of large gauge transformations for the $\mathbb{R}$ gauge field $A$.

On the physical boundary $\Sigma$ we want to impose the scale invariant boundary condition:

$$
F_t^{ij} = -\frac{i}{\mu}A^{ij},\ F = \mu_0 i A_t\ .
\tag{4.71}
$$

This correspond to the edge mode:

$$
\mathcal{L}_{\Sigma} = \frac{1}{4\pi}\left(\mu_0 A_t^2 - \frac{1}{\mu}A_{ij}^2\right)\ .
\tag{4.72}
$$

We can then compactify the slab and get the theory:

$$
\mathcal{L} = \frac{1}{2\pi}\left(\frac{\mu_0}{2}A_t^2 - \frac{1}{2\mu}A_{ij}^2 - \frac{i}{R}(\tilde{A}_{ij}(\partial_t A_{ij} - \partial_i\partial_j A_t) + \tilde{A}_t^{[ij]k}(\partial_i A_{jk} - \partial_j A_{ik}))\right)
\tag{4.73}
$$

If we then integrate out $\tilde{A}$, we get the constraints:

$$
\begin{aligned}
\partial_t A_{ij} - \partial_i\partial_j A_t &= 0 \\
\partial_i A_{jk} - \partial_j A_{ik} &= 0 \\
\oint dt A_t &\in \frac{1}{R}\mathbb{Z} \\
\int_{x_1^i}^{x_2^i} dx^i \oint dx^j A_{ij} &\in \frac{1}{R}\mathbb{Z}
\end{aligned}
\tag{4.74}
$$

which we can use to define:

$$
A_t = \partial_t\phi,\ A_{ij} = \partial_i\partial_j\phi,\ \phi \sim \phi + 2\pi R(n(x) + n(y) + n(z))\ .
\tag{4.75}
$$

The theory thus becomes:

$$
\mathcal{L}_\phi = \frac{1}{4\pi}\left(\mu_0(\partial_t\phi)^2 - \frac{1}{\mu}(\partial_i\partial_j\phi)^2\right)
\tag{4.76}
$$

which is exactly the $\phi$ theory described in (3.13).

### 4.3.5 Foliated edge modes

It is easy to see that the gapped edge mode for the foliated theory is:

$$\mathcal{L}_{\bar{\Sigma}} = \frac{iR}{2\pi}(\phi db + \Phi^i dB^i \wedge dx^i + \Phi^i b \wedge dx^i)$$

$$\delta\phi = -\frac{\lambda_0}{R}, \ \delta\Phi_i = -\frac{\lambda^i}{R}$$

(4.77)

where $\phi$ and $\Phi^i$ are compact real scalars of radius $R$. We can then compute the boundary conditions:

$$c = R(d\phi + \Phi^i \wedge dx^i)$$

$$C^i \wedge dx^i = Rd\Phi^i \wedge dx^i$$

$$db = 0$$

$$dB^i \wedge dx^i + b \wedge dx^i = 0$$

$$\oint b \in \frac{2\pi}{R}\mathbb{Z}$$

$$\int_{x_1^i}^{x_1^i} \oint B^i \wedge dx^i \in \frac{2\pi}{R}\mathbb{Z} \ .$$

(4.78)

We can then write a gapless edge modes requiring it to be scalar invariant and to transform well under discrete rotations, such gapless mode is:

$$\mathcal{L}_{\Sigma} = \frac{1}{4\pi}\left(\left(b + \sum_{ijk} d(B_{jk}^i dx^j \wedge dx^k)\right) \wedge *_3 \left(b + \sum_{ijk} d(B_{jk}^i dx^j \wedge dx^k)\right) \wedge dt\right)$$

$$+ \frac{1}{4\pi}\left((B^i - B^j) \wedge *_2(B^i - B^j) \wedge dx^i \wedge dx^j)\right) \ .$$

(4.79)

The theory after slab compactification becomes:

$$\mathcal{L}_\phi = \frac{1}{2\pi}\left(\frac{1}{2}\left(b + \sum_{ijk} dB^i\right) \wedge *_3 \left(b + \sum_{ijk} dB^i\right) \wedge dt + \frac{1}{2}(B^i - B^j) \wedge *_2(B^i - B^j) \wedge dx^i \wedge dx^j\right)$$

$$+ \frac{iR}{2\pi}(\phi db + \Phi^i dB^i \wedge dx^i + \Phi^i b \wedge dx^i) \ .$$

(4.80)

Integrating out the constraints $\Phi^i$ and performing the field redefinitions (4.63) we can show this theory to be dual to the $\phi$ theory. If we instead we integrate out the $\phi$ field, which locally sets $b = da_2$ we get another foliated version of a 2-form Maxwell gauge theory, which is dual to the tensor $\phi$-theory.

### 4.3.6 Boundary operators

Our aim is now to describe the bulk operators after slab compactification. Starting with the operators $\tilde{U}$ and $V$ in (4.67) and (4.64). The operator $\tilde{U}$ placed parallel to the $ij$ plane can be seen as the insertion of momentum operators: (3.15) in an interval:

$$\tilde{U}^{ij}_\alpha(x_1^k, x_2^k)[\Sigma^k] = \exp\left(i\alpha \int_{x_1^k}^{x_2^k} Q_{ij}(x^k)\right) , \tag{4.81}$$

acting on a line $V$ with end points on the boundaries and $\alpha = 2\pi nR$, corresponding to a local operator $\phi$.

The operator $\tilde{V}$ in (4.65) can be seen as the insertion of dipole operators (3.16):

$$\tilde{V}^{jk}_\alpha(x_1^i, x_2^i)[\sigma^i] = \exp\left(i\alpha \int_{x_1^i}^{x_2^i} Q(\gamma^{ij}(x^i), x^i)\right) \tag{4.82}$$

acting on the surface $U$ (4.66) with end-lines on the boundaries and $\alpha = 2\pi n/R$, corresponding to the a gauge invariant line operator (3.18) in the dual $\hat{A}$ theory. Note the quantized value of the charge for the operators ending at the boundary, given by the boundary conditions or (4.78).

## 4.4 $\hat{\phi}$-model

### 4.4.1 Exotic SymTFT

The Exotic-SymTFT that describes the $U(1)$ tensor symmetries in 3+1d is

$$\mathcal{L}_{sym_{\hat{\phi}}} = F^r_{ij}(\partial_t A^{ij} - \partial_k A_t^{k(ij)}) + F^t_{ij}(\partial_k A_r^{k(ij)} - \partial_r A^{ij}) + F(\partial_i \partial_j A^{ij}) + F_{k(ij)}(\partial_r A_t^{k(ij)} - \partial_t A_r^{k(ij)}) \tag{4.83}$$

where the exotic 1-form $\mathbb{R}$-gauge field $A$ is the $S_4$ representation $(\mathbf{3'}, \mathbf{3'}, \mathbf{2})$ with gauge transformations,

$$\delta A_t^{k(ij)} = \partial_t \lambda^{k(ij)},\ \delta A_r^{k(ij)} = \partial_r \lambda^{k(ij)},\ \delta A_{ij} = \partial_k \lambda^{k(ij)}. \tag{4.84}$$

The exotic 2-form $\mathbb{R}$-gauge field $F$ is in the $S_4$ representation $(\mathbf{1}, \mathbf{3'}, \mathbf{3'}, \mathbf{2})$ with gauge transformation,

$$\begin{aligned}
\delta F &= \partial_t \chi_r - \partial_r \chi_t \\
\delta F^t_{ij} &= \partial_t \chi^{ij} - \partial_i \partial_j \chi_t \\
\delta F^r_{ij} &= \partial_r \chi^{ij} - \partial_i \partial_j \chi_r \\
\delta F_{[ij]k} &= \partial_i \chi_{jk} - \partial_j \chi_{ik}
\end{aligned} \tag{4.85}$$

where $\chi$ is an exotic gauge parameter in $S_4$ representation $(\mathbf{1}, \mathbf{1}, \mathbf{3'})$. The equation of motions for this theory are

$$\partial_t A^{ij} - \partial_k A_t^{k(ij)} = 0$$
$$\partial_k A_r^{k(ij)} - \partial_r A^{ij} = 0$$
$$\partial_i \partial_j A^{ij} = 0$$
$$\partial_r A_t^{k(ij)} - \partial_t A_r^{k(ij)} = 0 \tag{4.86}$$
$$\partial_t F_{k(ij)} - \partial_k F_{ij}^t = 0$$
$$\partial_r F_{k(ij)} - \partial_k F_{ij}^r = 0$$
$$\partial_t F_{ij}^r - \partial_t F_{ij}^t + \partial_i \partial_j F = 0 \ .$$

### 4.4.2 Foliated SymTFT

The dual, foliated theory describing the same symmetries is

$$\mathcal{L}_{sym_{\hat{\phi}}} = \frac{1}{2\pi}(c \wedge db + C^i \wedge dB^i \wedge dx^i + b \wedge C^i \wedge dx^i) \tag{4.87}$$

where $b$ and $c$ are 2-form $\mathbb{R}$ gauge fields, $B^i$ is a 1-form $\mathbb{R}$ gauge field, and $C^i$ is a 2-form foliated $\mathbb{R}$ gauge field. The related gauge transformations are,

$$\delta b = d\chi_1$$
$$\delta c = d\lambda_1 - \sum_i \lambda_1^i dx^i$$
$$\delta B^i = d\chi_0 - \chi_1 \tag{4.88}$$
$$\delta C^i = d\lambda_1^i$$

and the equations of motion are:

$$db = 0$$
$$dB^i \wedge dx^i - b \wedge dx^i = 0$$
$$dc + \sum_i C^i \wedge dx^i = 0 \tag{4.89}$$
$$dC^i \wedge dx^i = 0 \ .$$

### 4.4.3 Gauge invariant operators

We can once again study the gauge invariant operators for these theories. We will present the operators for the dual theories at the same time. Their equivalence is provided by the

identification,

$$A^{ij} = B^i_k - B^j_k, \, A^{k(ij)}_t = B^i_t - B^j_t$$

$$F = c_{rt}, F_{[ij]k} = C^k_{ij} + \partial_k c_{ij} \tag{4.90}$$

$$F^t_{ij} = C^i_{tj} + \partial_i c_{tj} + \partial_t c_{ij} \, .$$

We start with two dual lines:

$$U^k_\alpha(x^i, x^j)[\gamma] = \exp\left( i\alpha \oint A^{k(ij)}_t dt + A^{k(ij)}_r dr + A^{ij} dx^k \right)$$

$$= \exp\left( i\alpha \oint B^i - B^j \right) \, . \tag{4.91}$$

We have a strip operator:

$$\tilde{U}^{k(ij)}_\alpha(x^k_1, x^k_2)[\sigma^k] = \exp\left( i\alpha \int_{x^k_1}^{x^k_2} \oint_\gamma \partial_k A^{jk} dx^i + (\partial_j A^{ij} + \partial_k A^{ik}) dx^j \right.$$

$$\left. + \int_{x^k_1}^{x^k_2} \oint_\gamma \partial_k A^{i(jk)} dt + \partial_r A^{i(jk)} dr \right) \tag{4.92}$$

$$= \exp\left( i\alpha \int_{x^k_1}^{x^k_2} \oint_\gamma b + dB^i + dB^j \right) \, .$$

We then have the conjugate operators; two dual slabs:

$$\tilde{V}^{[ij]k}_\alpha(x^k_1, x^k_2)[\Sigma^k] = \exp\left( i\alpha \int_{x^k_1}^{x^k_2} \oint_\Gamma F_{[ij]k} dx^i dx^j + F^t_{ik} dt dx^i + F^t_{jk} dt dx^j \right.$$

$$\left. + \int_{x^k_1}^{x^k_2} \oint_\Gamma + F^r_{ik} dr dx^i + F^r_{jk} dr dx^j + \partial_k F dr dt \right) \tag{4.93}$$

$$= \exp\left( i\alpha \int_{x^k_1}^{x^k_2} \oint_\Gamma C^k dx^k + dc \right)$$

where the index $k$ in $\tilde{V}^{[ij]k}$ has been added to make the $S_4$ representation manifest.

The two dual surfaces:

$$V(x, y, z)_\alpha = \exp\left( i\alpha \oint F dr dt \right)$$

$$= \exp\left( i\alpha \oint c \right) \, . \tag{4.94}$$

We can describe the linking of these objects:

$$\langle U^i_\alpha(x^j, x^k)[\gamma], \tilde{V}^{[ij]k}_\beta(x^k_1, x^k_2)[\Sigma^k] \rangle = e^{2\pi i \alpha \beta \mathrm{Link}_k(\gamma, \Sigma^k)}$$

$$\langle \tilde{U}^{k(ij)}_\alpha(x^k_1, x^k_2)[\sigma^k], V_\beta(x, y, z) \rangle = e^{2\pi i \alpha \beta \mathrm{Link}_k(\sigma^k, \Gamma_{rt})} \tag{4.95}$$

where $\Gamma_{rt}$ is the $rt$ plane.

### 4.4.4 Exotic edge modes

We can now write the gapped edge mode that will make this theory gauge invariant:

$$\mathcal{L}_{\bar{\Sigma}} = \frac{i}{2\pi R}\hat{A}_{ij}(\partial_t A^{ij} - \partial_k A^{k(ij)}) + \hat{A}_t(\partial_i\partial_j A^{ij})$$
$$\delta\hat{A}_t = -\chi_t R, \ \delta\hat{A}_{ij} = -\chi_{ij}R \tag{4.96}$$

Where $\hat{A}$ is an exotic 1-form $U(1)$-gauge field in representation $(\mathbf{1}, \mathbf{3}')$. Once again, this is the Lagrangian of the X-cube model whit an $\mathbb{R}$ gauge field. We can then set a scale invariant boundary condition on $\Sigma$:

$$\mathcal{L}_{\Sigma} = \frac{1}{4\pi}\left(\frac{\mu}{6}(A_t^{k(ij)})^2 - \frac{\mu_0}{2}(A^{ij})^2\right) \tag{4.97}$$

Once again, after slab compactification we can get the lagrangian for the $\hat{\phi}$-model(3.19).

### 4.4.5 Foliated edge modes

We can instead add a gapped edge mode to the foliated theory:

$$\mathcal{L}_{\bar{\Sigma}} = \frac{i}{2\pi R}(v \wedge db + V^i \wedge dB^i \wedge dx^i + V^i \wedge b \wedge dx^i)$$
$$\delta v = -\frac{\lambda_i}{R}, \ \delta V^i = -\frac{\lambda_1^i}{R} \tag{4.98}$$

Where $v$ and $V^i$ are a $U(1)$ 1-form gauge field, setting the boundary conditions:

$$c = RdV + V^i \wedge dx^i$$
$$C^i \wedge dx^i = dV^i \wedge dx^i$$
$$db = 0$$
$$dB^i \wedge dx^i + b \wedge dx^i = 0 \tag{4.99}$$
$$\oint b \in \frac{2\pi}{R}\mathbb{Z}$$
$$\int_{x_1^i}^{x_2^i} \oint B^i \wedge dx^i \in \frac{2\pi}{R}\mathbb{Z} \ .$$

The scale invariant edge mode is:

$$\mathcal{L}_{\Sigma} = \frac{1}{4\pi}(B^i - B^j) *_2 (B^i - B^j) \wedge dx^i \wedge dx^j \ . \tag{4.100}$$

Once again, after slab compactification we can get the foliated version of the $\hat{\phi}$ theory (3.19):

$$\mathcal{L}_{\hat{\phi}} = \frac{1}{2\pi}\left(\frac{1}{2}(B^i - B^j) *_2 (B^i - B^j) \wedge dx^i \wedge dx^j - (\frac{i}{R}(v \wedge db + V^i \wedge dB^i \wedge dx^i + V^i \wedge b \wedge dx^i)\right) \tag{4.101}$$

If we integrate $v$ out we locally get $b = da_2$. By plugging this back into the action we get another foliated version of Maxwell theory for the 2-form gauge field $a_2$, which is dual to the $\hat{\phi}$-model.

### 4.4.6 Boundary operators

In this case the operators $U$ and $\tilde{V}$ can be used to build both the symmetry and the charged operators. When the line $U$ ends on the boundary it creates the operator $\hat{\phi}$ charged under the defect (3.21):

$$\tilde{V}_\alpha^{[ij]k}(x_1^k, x_2^k) = \exp\left(i\alpha \int_{x_1^k}^{x_2^k} Q^{[ij]}\right) , \tag{4.102}$$

where $\tilde{V}$ is inserted on a slab on the $ij$ plane.

When the strip $\tilde{V}$ ends on the boundary it creates the operator $W$ in (3.24), and the symmetry operator is the line $U$

$$U_\alpha^k(x^i, x^j) = \exp\left(i\alpha Q^k(x^i, x^j)\right) , \tag{4.103}$$

where $U$ is inserted on a line along the $x^k$ direction. Once again, the boundary conditions (4.99) impose the quantization conditions for the tensor charge.

## 5 Conclusion and outlook

In this paper we have studied gapless exotic-foliated dual models in various dimensions via the SymTFT *Mille-feuille* construction for continuous subsystem symmetries. We provided the explicit foliated free lagrangian model dual to exotic models like XY-plaquette, XYZ-cube, $\phi$ and $\hat{\phi}$ models. These models appear as spontaneous symmetry breaking of the subsystem symmetries whose structure we described in terms of the continuous SymTFT. We also extract all possible topological manipulation of these models via the SymTFT construction and its topological boundary conditions. This often coincides with changing the periodicity of the scalars in the exotic models, and has a counterpart in the foliated ones.

It would be very interesting to extend our approach and provide the systematic tool that realizes the SymTFT both in the exotic and foliated case, only by inputting the representation of discrete subgroups of rotation symmetry. The SymTFT Mille-feuille construction will naturally provide its set of topological boundary conditions as well as its symmetry breaking, or scale invariant, one. Implementing this will give rise to the tool to construct very general free theories with subsystem symmetries and fractonic excitations, if not classify them.

## Acknowledgement

The authors want to thank Riccardo Argurio for inspiring discussion. The work of FA and FB is supported in part by the Italian MUR Departments of Excellence grant 2023-2027 "Quantum Frontiers". The work of SM is supported by the University of Padua under the 2023

STARS Grants@Unipd programme (GENSYMSTR – Generalized Symmetries from Strings and Branes) and in part by the Italian MUR Departments of Excellence grant 2023-2027 "Quantum Frontiers". The research of FA was supported in part by grant NSF PHY-2309135 to the Kavli Institute for Theoretical Physics (KITP).

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
