# Peer review of "SymTFT construction of gapless exotic-foliated dual models"

_SciPost Physics_

## Round 2 · Referee Report · Anonymous (Referee 1) · 2025-11-3

Report

The symmetry topological field theory (SymTFT) is an important tool to study dualities and symmetries. It has been generalized to describe the discrete subsystem symmetries and ordinary continuous symmetries. The authors extend this idea to construct SymTFT for continuous subsystem symmetries. Their focus is to reproduce gapless model with continuous subsystem symmetry, which is given by choosing gapped and symmetry-breaking boundary conditions of the SymTFT. They also discuss the duality between two formulations of the bulk SymTFT, using foliated field theory or exotic field theory, which is also a natural generalization from the discrete cases.

This work is a natural generalization in this field and will help the further study of gapless models, or spontaneous symmetry breaking phases, with subsystem symmetries. However, I hope the authors can clarify the following points in the requested changes.

Requested changes

The SymTFT gains its power at separating the symmetrical and dynamical data at different boundaries, and is universal for different models if the symmetry is given. In this paper the authors only studied one specific choice of boundary condition that is used to reproduce the models reviewed in section 3. This may diminish this article's significance and impact. Therefore, I hope the authors can address the following questions:

  1. Is it easy to study or classify different topological boundary conditions (e.g. Dirichlet or Neumann) using Mille-feuille method?
  2. What is the bulk operator that implement the change of boundary conditions? (Is it the condensation operator?)
  3. What is the dual models after changing a different boundary conditions?
  4. Usually the condensation operators obeys noninvertible fusion rules. What is the fusion rules in this case?
  5. Does the Mille-feuille method help us understand the UV-IR mixing related to subsystem symmetry? For example, after Eq (4.20), the authors claimed "For instance, we required scale invariance when constructing the physical boundary, not allowing for some terms that, due to UV/IR mixing, will contribute to the energy at leading order. As a consequence, the energy of our ground state is shifted. " Can you give the details of this?

At least, I hope the authors can clarify the questions in the example related to XY-plaquette models.

Recommendation

Ask for major revision

---

## Round 2 · Referee Report · Anonymous (Referee 2) · 2025-12-8

Strengths

The construction of foliated/exotic SymTFTs for gapless theories is quite new to the field.

Weaknesses

Some of the explanations are not very clear, and several notational conventions would benefit from a more explicit description.

Report

The paper studies the SymTFTs associated with continuous subsystem symmetries. The resulting SymTFTs take the form of foliated (or exotic) field theories. The authors use four representative models as illustrative examples: the XY-plaquette model, the XYZ-cube model, the $\phi$-model, and the $\hat{\phi}$-model, following mainly Refs.\,[48], [49], and [51].
We note that what the authors call the XYZ-cube model'' is referred to as theXY-cube model'' in the existing literature.

The main contribution of the paper is to introduce some foliated/exotic field theories as the bulk theories, impose gapped boundary conditions on the symmetry boundary and gapless boundary conditions on the physical boundary, and show that the resulting SymTFTs correspond to the XY-plaquette model, the XYZ-cube model, the $\phi$-model, and the $\hat{\phi}$-model.

I find the results presented in the manuscript interesting and recommend the publication of the paper after some revisions. I have several questions for the authors:

  • The method used to impose the gapped boundary condition leads to a particular choice of gapped boundary. Is there a way to obtain alternative gapped boundary conditions, for example by manipulating the bulk theory using dualities or symmetry actions? If the resulting theory is related to the original one (e.g.\ via gauging), can the authors comment on this?
  • Determining topological boundary conditions of a foliated field theory is a crucial but challenging problem. For the bulk foliated theory considered in the paper, do the authors have ideas on how to determine at least some classes of gapped boundary conditions?
  • Since all four models discussed arise from lattice constructions, could the authors provide comments on the implications of their SymTFT analysis for the corresponding lattice models?

In addition, some of the notation appears to be conventional to the paper and would benefit from clearer explanation to improve readability.

The following are some minor issues:

  • Page 5, Eq.(2.1): $M_{d}+1$'' in the integration domain should be$M_{d+1}$''.
  • Page 5, line after Eq.(2.1): $b_{d-p-1}$ and $c_{p+1}$ are $\mathbb{R}$-valued gauge field'' should begauge fields''.
  • Eq.(3.19) should read
    $$ \mathcal{L}=\frac{\hat{\mu}{0}}{12}(\partial -\frac{\hat{\mu}}{4}(\partial_{k}\hat{\phi}^{k(ij)})^{2}. $$}\hat{\phi}^{i(jk)})^{2
    The description involving $\hat{\phi}$ uses many indices and conventions; a clearer explanation of these notations would help the reader.
  • Page 18, line before Eq.(4.34): See 3'' should readSee Figure 3''.
  • Page 20, after Section title 4.2.1: XY-cube'' should beXY-plaquette model''.

Requested changes

See report

Recommendation

Ask for minor revision

---

## Editorial Decision

awaiting_resubmission